# Brief communication: PICOP, a new ocean melt parameterization under ice shelves combining PICO and a plume model

Tyler Pelle[1], Mathieu Morlighem[1], and Johannes H. Bondzio[1]

[1]University of California, Irvine, Department of Earth System Science, Irvine, CA 92697-3100, USA

**Correspondence:** Tyler Pelle (tpelle@uci.edu)

**Abstract.** Basal melting at the bottom of Antarctic ice shelves is a major control on glacier dynamics, as it modulates the amount of buttressing that floating ice shelves exert onto the ice streams feeding them. Three-dimensional ocean circulation numerical models provide reliable estimates of basal melt rates but remain too computationally expensive for century scale projections. Ice sheet modelers therefore routinely rely on simplified parameterizations either based on ice shelf depth or on more sophisticated box models. However, existing parameterizations do not accurately resolve the complex spatial patterns of sub-shelf melt rates that have been observed over Antarctica's ice shelves, especially in the vicinity of the grounding line, where basal melting is one of the primary drivers of grounding line migration. In this study, we couple the Potsdam Ice-shelf Cavity mOdel (PICO, Reese et al., 2018) to a buoyant plume melt rate parameterization (Lazeroms et al., 2018) to create PICOP, a novel basal melt rate parameterization that is easy to implement in transient ice sheet numerical models and produces a melt rate field that is in excellent agreement with the spatial distribution and magnitude of observations for several ocean basins. We test PICOP on the Amundsen Sea sector of West Antarctica, Totten and Moscow University ice shelves in Eastern Antarctica, and the Filchner-Ronne Ice Shelf and compare the results to PICO. We find that PICOP is able to reproduce inferred high melt rates beneath Pine Island, Thwaites, and Totten glaciers (on the order of 100 m/yr) and removes the "banding" pattern observed in melt rates produced by PICO over the Filchner-Ronne Ice Shelf. PICOP resolves many of the issues contemporary basal melt rate parameterizations face and is therefore a valuable tool for those looking to make future projections of Antarctic glaciers.

## 1 Introduction

Glaciers around the periphery of the Antarctic Ice Sheet (AIS) have undergone dynamic changes due to the spreading of warm modified Circumpolar Deep Water (mCDW) onto the continental shelf and, sometimes, into sub-ice shelf cavities (e.g., Jacobs et al., 2011; Pritchard et al., 2012). This process drives enhanced basal melting, which has the potential to reduce the buttressing effect that ice shelves exert on grounded ice upstream (e.g., Rignot and Jacobs, 2002). This spreading of mCDW is expected to increase along sectors of the periphery of the AIS due to the poleward intensification of the Southern Hemisphere

westerly winds (Dinniman et al., 2012). As such, accurately parameterizing these basal melt rates is necessary in making future projections of the AIS due to the large computational cost of two way ice-ocean model coupling. Many early basal melt rate parameterizations (i.e., parameterizations based on the local heat flux at the ice-ocean interface (DeConto and Pollard, 2016; Beckmann and Goosse, 2003) or on basal slopes (Little et al., 2012)) do not accurately capture the impact of ocean circulation

within sub-shelf cavities, which is a key control of basal melting. Two of the most recently published melt parameterizations that resolve sub-shelf ocean circulation are the Potsdam Ice-shelf Cavity mOdel (PICO, Reese et al., 2018) and one based on the physics of buoyant melt water plumes (plume model, Lazeroms et al., 2018). Although both parameterizations are novel in their own regards, melt rates calculated by PICO suffer from unrealistic "banding" as a product of its box model approach and remain too low near grounding lines. In addition, the plume model requires complete sub-shelf ocean temperature and

salinity fields as inputs, and has not been adapted to use in transient model runs. We overcome these limitations by combining both PICO and the plume model to form PICOP: we rely on PICO's box model to reconstruct the temperature and salinity fields beneath ice shelves based on far field ocean properties and then use this reconstruction to drive the plume model, which calculates the basal melt rate field. In this brief communication, we describe the physics used to derive PICOP and compare melt rates produced by PICO and PICOP to observations by Rignot et al. (2013) in three basins of varying oceanic conditions

and geometry.

## 2 Methods

### 2.1 PICO

PICO is a two-dimensional sub-shelf melt rate parameterization that simulates vertical overturning in sub-shelf cavities and is used here to produce ambient ocean temperature and salinity fields (Reese et al., 2018). Inputs for PICO are the basin-averaged

ocean temperature $T$ and salinity $S$ and sub-shelf ocean circulation is driven by the *ice pump* mechanism (Lewis and Perkin, 1986). Individual mesh elements or grid cells within the model domain are assigned a box number based on their relative distance from both the grounding line and ice front. In general, PICO solves for the transport of heat and salt between boxes in contact with the base of the ice shelf, starting at the grounding line and ending at the ice front (boxes $B_k$ for k = $\{1, \ldots, n\}$, where $n$ is typically less than or equal to 5). After simplification and assuming steady state conditions, the balance of heat and

salt in all boxes along the base of the ice shelf can be written as:

$$
\begin{aligned}
q\left(T_{k-1} - T_k\right) - A_k m_k \frac{\rho_i}{\rho_w} \frac{L}{c_p} &= 0 \\
q\left(S_{k-1} - S_k\right) - A_k m_k S_k &= 0.
\end{aligned}
\tag{1}
$$

Using a simplified formulation of the 3-equation melt model by Holland and Jenkins (1999), the transport equations can be solved for salinity $S_k$ and temperature $T_k$ in box $B_k$, and are dependent on the local pressure $p_k$, the box area $A_k$, and the temperature $T_{k-1}$ and salinity $S_{k-1}$ of the upstream box $B_{k-1}$. The strength of the overturning circulation, $q$, is calculated once

per time-step in box $B_1$ from the density difference between the far field and grounding line water masses:

$$
q = C(\rho_0 - \rho_1).
\tag{2}
$$

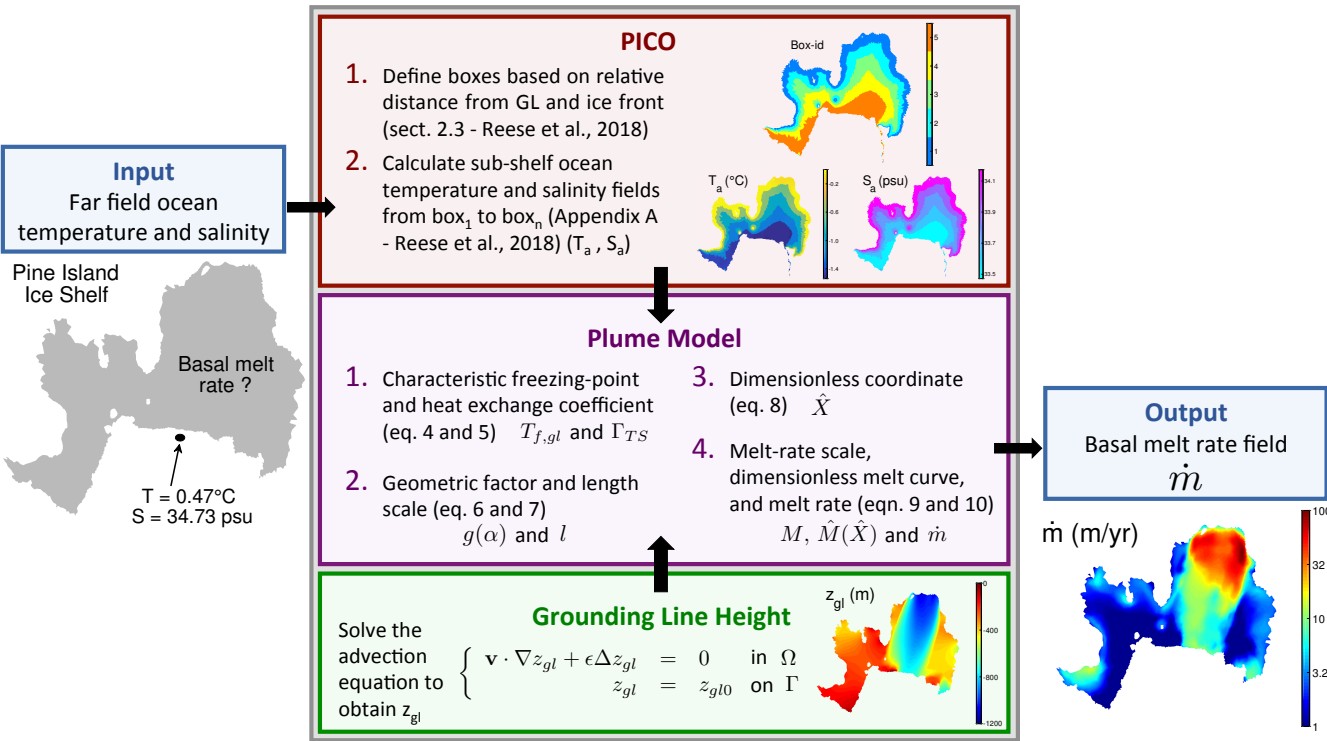

**Figure 1.** Schematic diagram of PICOP with example data displayed for the Pine Island ice shelf of West Antarctica. The inputs into the parameterization are the basin averaged ocean temperature ($^{\circ}$C) and salinity (psu), which are first fed into PICO (red box). PICO uses these inputs to calculate the sub-shelf ambient ocean temperature and salinity fields, which are then used in the plume model (purple box). In addition, the grounding line height is calculated at this time by solving the advection problem defined in the green box. Once these three fields are fed into the plume model, the basal melt rate field is computed according to the steps outlined in the purple box.

Here, we do not use PICO's melt rate parameterization but only use the sub-shelf temperature and salinity fields to drive the plume model (Fig. 1). All constants and external parameters referenced in this paper are summarized in Table 1. For a full derivation of PICO, see Reese et al. (2018).

## 2.2 Plume model

5 The plume model is a basal melt rate parameterization based on the theory of buoyant melt water plumes that travel upward along the base of the ice shelf from the grounding line to the location where the plume loses buoyancy. The two-dimensional formulation from Lazeroms et al. (2018) is adapted from the one-dimensional plume model developed by Jenkins (1991) for a plume traveling in direction $X$ in an ocean with ambient temperature $T_a$ and salinity $S_a$ (provided by PICO). We begin by defining the grounding line depth, $z_{gl}$, over the entire ice shelf, as it is necessary to determine where individual plumes

originate in order to employ this parameterization. As a first approximation, we solve an advection equation:

$$\begin{cases} \mathbf{v} \cdot \nabla z_{gl} + \epsilon \Delta z_{gl} & = & 0 & \text{in } \Omega \\ z_{gl} & = & z_{gl0} & \text{on } \Gamma \end{cases} \tag{3}$$

where $z_{gl0}$ is the grounding line height defined at the grounding line $\Gamma$, $\Omega$ is the ice shelf, and as a first approximation, $\mathbf{v}$ is the modeled, depth-averaged ice velocity. Note that $\epsilon$ is a small diffusion coefficient introduced to minimize noise and to
provide numerical stability. We attempted using other advection schemes, for example based on basal slopes, but the level of noise made these approaches unpractical. As such, we make the assumption that the source of individual melt water plumes coincides with the direction of ice velocity. That is, for any given point $x$ on the base of an ice shelf, the grounding line height $z_{gl}(x)$ (i.e. the depth at which the plume originates) associated with that point can be found by following an ice flowline upstream of $x$ to $\Gamma$. Note that this does not specify the path the plume takes from $z_{gl}(x)$ to $x$. The path the plume traverses
is a product of the ice-shelf basal slopes, which is acted on by changes in ice shelf thickness along the plume's trajectory. If areas of ice convergence and divergence on a shelf are neglected, we generally expect for ice-shelf thickness to decrease as we move from the grounding line to the ice front. Since melt water plumes are driven by buoyancy, it is then reasonable to assume that for small ice shelves, the average trajectory of a plume would be from the grounding line to the ice front. As such, using the ice velocity in the advection scheme to approximate the depth at which the plume originates is not an unreasonable
assumption as a first approximation. For larger ice shelves however, sub-shelf flow is affected by different mechanisms that cannot be captured by a simplified parameterization, such as polynya variability.

In a second step, we correct $z_{gl}$ such that, if $z_{gl}$ is greater than the height of the base of the ice shelf, $z_b$, then we set $z_{gl} = z_b$. Compared to the algorithm used to determine $z_{gl}$ in Lazeroms et al. (2018), advecting grounding line heights is computationally more efficient for higher resolution model runs because we do not have to search for multiple possible plume sources at every
point within a given ice shelf.

Now that $z_{gl}$ is defined, we continue by computing both the characteristic freezing point $T_{f,gl}$ and the effective heat exchange coefficient $\Gamma_{TS}$ as follows:

$$T_{f,gl} = \lambda_1 S_a + \lambda_2 + \lambda_3 z_{gl} \tag{4}$$

$$\Gamma_{TS} = \Gamma_T \left( \gamma_1 + \gamma_2 \frac{T_a - T_{f,gl}}{\lambda_3} \times \frac{E_0 \sin \alpha}{C_d^{1/2} \Gamma_{TS_0} + E_0 \sin \alpha} \right). \tag{5}$$

A geometric scaling factor $g(\alpha)$ and length scale $l$ are defined in order to give the plume model the proper geometry dependence and scaling according to the distance traveled along the plume path. The scaling factor and length scale are computed as follows:

$$g(\alpha) = \left( \frac{\sin \alpha}{C_d + E_0 \sin \alpha} \right)^{1/2} \left( \frac{E_0 \sin \alpha}{C_d^{1/2} \Gamma_{TS} + E_0 \sin \alpha} \right)^{1/2} \left( \frac{E_0 \sin \alpha}{C_d^{1/2} \Gamma_{TS} + E_0 \sin \alpha} \right) \tag{6}$$

$$l = \frac{T_a - T_{f,gl}}{\lambda_3} \times \frac{x_0 C_d^{1/2} \Gamma_{TS} + E_0 \sin \alpha}{x_0 \left( C_d^{1/2} \Gamma_{TS} + E_0 \sin \alpha \right)}. \tag{7}$$

The dimensionless scale factor $x_0$ used in the second term of $l$ defines the transition-point between melting and refreezing and is constant for all model results. For a complete explanation of the individual terms that make up these two factors, see section 2.2 of Lazeroms et al. (2018).

The length scale is then used in the computation of the dimensionless coordinate, $\hat{X}$:

$$\hat{X} = \frac{z_b - z_{gl}}{l}. \tag{8}$$

Note that $\hat{X} = 0$ corresponds to the position of the grounding line and $\hat{X} = 0.56$ is the aforementioned transition point, but $\hat{X} = 1$ does not necessarily correspond to the position of the calving front due to the dependence of $\hat{X}$ on $l$. In order to ensure valid values of $\hat{X}$, we set a lower bound for the ambient ocean temperature: $T_a \geq \lambda_1 S_a + \lambda_2$. The melt rate $\dot{m}$ is then calculated as

$$\dot{m} = \hat{M}(\hat{X}) \times M, \tag{9}$$

where $\hat{M}(\hat{X})$ is a dimensionless melt curve defined in Lazeroms et al. (2018) and $M$ is defined as

$$M = M_0 \times g(\alpha) \times \left(T_a - T_f\left(S_a, z_{gl}\right)\right)^2. \tag{10}$$

For a full derivation of the buoyant plume model used in PICOP, see Lazeroms et al. (2018).

## 3   Results and Discussion

We evaluate PICOP using geometry from Bedmap2 (Fretwell et al., 2013) and far field ocean temperature and salinity values averaged at the depth of the continental shelf between 1975 to 2012 (Reese et al., 2018; Schmidtko et al., 2014). Here, we compare the modeled basal melt rates calculated by PICO and PICOP to melt rates inferred from conservation of mass and satellite interferometry (Rignot et al., 2013), that we refer to as "observations". Additionally, we compare the modeled basal melt rate field of select ice shelves to in-situ observations and regional modeling studies. We focus on three regions: the Amundsen Sea sector of the West Antarctic Ice Sheet, the Totten and Moscow University ice shelves of the East Antarctic Ice Sheet, and the Filchner-Ronne Ice Shelf (FRIS). Model inputs for these basins are $(0.47°C, 34.73$ psu$)$, $(−0.73°C, 34.73$ psu$)$, and $(−1.76°C, 34.82$ psu$)$, respectively.

The spatial distribution of melt rates produced by PICOP is in significantly better agreement with observations compared to PICO, especially in the vicinity of the grounding line where accurate melt rates are needed in order to correctly capture the glacier's grounding line dynamics. In Fig. 2, we see that modeled melt rates produced by PICOP reach approximately 100 m/yr and 70 m/yr near the grounding line of Pine Island and Thwaites glaciers, respectively, as compared to approximately 20 m/yr by PICO. These high melt rates are a product of the deeply entrenched bed that both Pine Island and Thwaites glaciers are grounded to. These bed depths are advected with the modeled ice velocity when $z_{gl}$ is solved for, leading to high melt rates that better match observations. Melt rates modeled by Dutrieux et al. (2013), constrained by high resolution satellite and airborne observations of ice surface velocity and elevation, show melt rates on the order of 100 m/yr near Pine Island glacier's

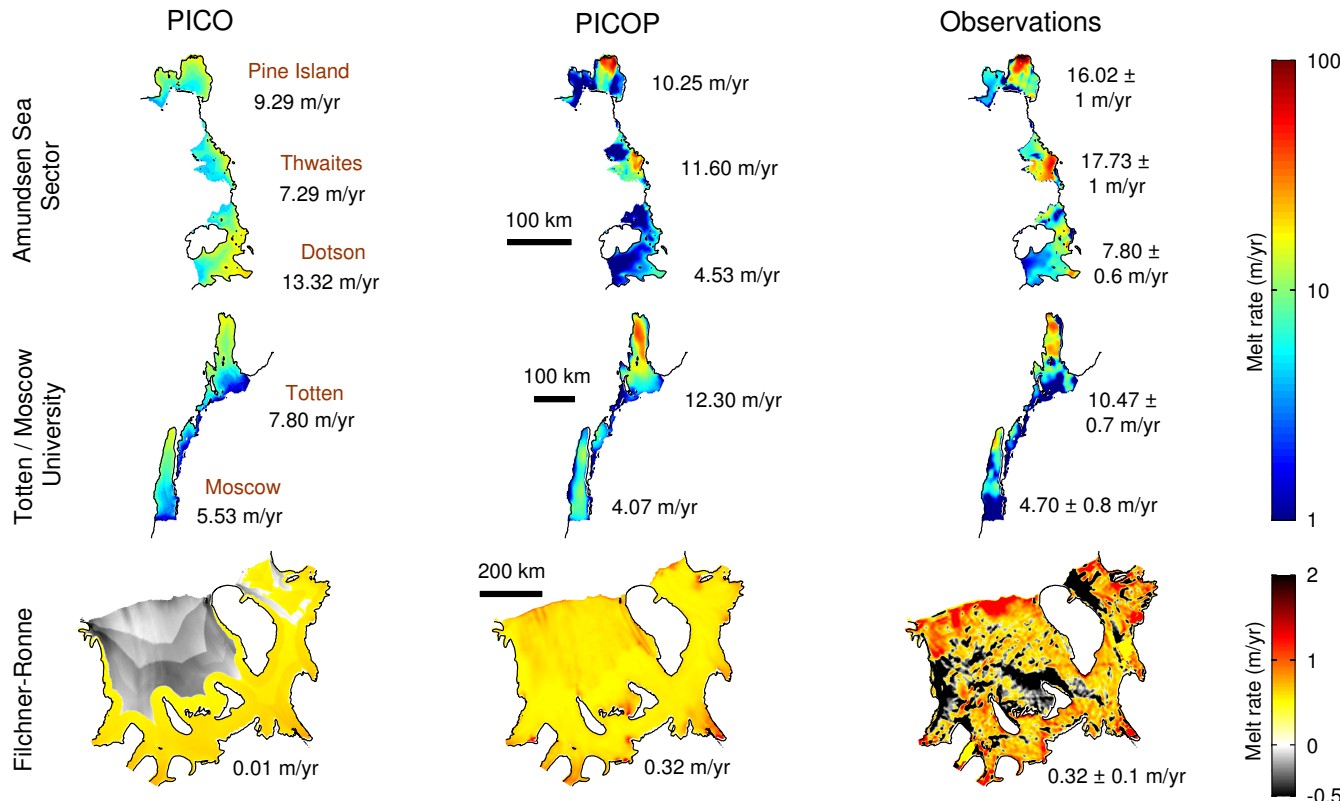

**Figure 2.** Modeled (PICO and PICOP) and observed (Rignot et al., 2013) melt rates (m/yr) are displayed for the Amundsen Sea sector of West Antarctica (including Pine Island, Thwaites, and Dotson ice shelves), Totten and Moscow University ice shelves of East Antarctica, and the FRIS. Note that the upper colorbar (Amundsen Sea sector, Totten, and Moscow University) is in log-form while the lower colorbar (FRIS) is linear. Numerical values under PICO and PICOP are area weighted mean melt rates. The observed annual mean melt rate is displayed under the observed melt rate panel.

grounding line and 30 m/yr a short 20 km downstream. This sharp gradient in the melt rate field was reproduced by PICOP and will certainly have a major impact on the ice dynamics of this glacier.

A similar situation occurs under Totten ice shelf; melt rates modeled by PICOP reach a maximum of about 50 m/yr, while those from PICO reach a maximum of approximately 20 m/yr. Simulated melt rates by Gwyther et al. (2014) show a similar
5  pattern of melt, with basal melt rates of approximately 50 m/yr computed near the upstream most portion of both Totten and Moscow University's grounding lines. Modeling these high melt rates is especially important in this region of Totten's grounding line, as complex grounding line retreat has been observed over the past 17 years and has been found to be strongly sensitive to changes in ocean temperature (Li et al., 2015). Over the FRIS, the inherent geometry dependence of PICOP reduced the "banding" that modeled melt rates from PICO displayed. This is a significant improvement because as can be
10  seen in Fig. 2, there is a very sharp gradient in the melt rate field computed by PICO over the FRIS that would lead to

unrealistic ice-shelf dynamics in transient model runs. PICOP produces a smooth transition from high to low melt rates that better matches observations. Shelf-wide basal melt rate fields computed by three dimensional ocean - ice shelf coupled models (e.g., Timmermann et al., 2012) show maximum melting (4.5 - 7 m/yr) near the deepest sectors of the grounding line of Ronne glacier, agreeing well with PICOP. Site specific observations (e.g., Jenkins et al., 2010) show a decrease in basal melting to less than 1 m/yr near the Kroff ice rise, which is also reproduced by PICOP.

In all three basins, area-weighted mean melt rates calculated with PICOP show better agreement with Rignot et al. (2013). The values reported in figure two corresponding to PICO differ from those used in figure 5 of Reese et al. (2018) because we model these basins using a significantly higher mesh resolution (minimum element size of 500 m, maximum of 10 km). By modeling Totten, Pine Island, and Thwaites ice shelves with a coarse mesh, only two boxes were defined for these smaller shelves in Reese et al. (2018), and thus, a larger proportion of the ice shelf was modeled as the grounding line box. Melt rates computed in this box are the highest across the shelf because no heat has been lost from the ocean water by the addition of cold melt water, leading to higher mean melt rates when compared to those displayed in figure 2. By using a finer mesh to evaluate PICOP, we are able to capture the fine details of the melt pattern, which are key in predicting the evolution of grounding line dynamics, as well as maintain shelf-averaged melt rates that are in relatively good agreement with observations. The mean melt rates for Pine Island and Thwaites ice shelves are underestimated (10.25 m/yr and 11.60 m/yr, respectively), as calculated melt rates are too low away from the vicinity of the grounding line. In addition, the mean melt rate for Totten ice shelf is slightly overestimated (12.30 m/yr) when modeled with PICOP due to the strong grounding line advection used to compute $z_{gl}$ in this region. Over the FRIS, PICOP models a shelf-mean melt rate that is in better agreement with observations than PICO, because PICOP produces melt further downstream of the grounding line as a result of its geometry dependence. In this sector of the ice shelf, PICO primarily computes refreezing ($\dot{m} < 0$), which drives the mean melt rate down to 0.01 m/yr.

While PICOP resolves many of the issues displayed in contemporary sub-shelf mate rate parameterizations, it is limited by the assumptions that were made when both PICO and the plume model were originally derived (see Reese et al. (2018) and Lazeroms et al. (2018)). In addition, when computing $z_{gl}$, we assumed that the depth of the plume origin at any point on an ice shelf could be found by following the flow of velocity upstream to the grounding line. Although a good first approximation, we expect this assumption to fail in zones of complex basal geometry (i.e. areas of convergent and divergent ice flow) that would lead melt-water plumes to follow more convoluted paths. We also expect this assumption to fail in large sub-shelf cavities, such as under the FRIS or Ross ice shelf, where plume paths are influenced by processes not captured by this parameterization (i.e. sea ice and polynya variability, the Coriolis Effect that produces a clockwise sub-shelf ocean circulation, and tides). Finally, PICOP does not model refreezing well in cold basins due to the lower limit imposed on the ambient ocean temperature. The ocean temperature output from PICO in cold basins (i.e. the FRIS and Ross ice shelf) falls below this lower bound, especially in the vicinity of the ice front, where the coldest ocean temperatures are modeled. As such, melt rates computed in the coldest cavities might be over estimated and cannot be further improved unless this constraint is relaxed, as discussed in Appendix A of Lazeroms et al. (2018). This is exemplified in the modeled basal melt rates produced by PICOP in Fig. 2. Observations show patches of refreezing under the FRIS that are not resolved by PICOP as a result of this lower temperature bound. Yet,

PICOP remains an accurate and computationally efficient melt rate parameterization that can be easily implemented into high resolution, transient ice sheet numerical models.

## 4   Conclusions

Here, we presented a new basal melt rate parameterization that is a combination of both PICO and a plume model. By utilizing PICO to resolve the sub-shelf ocean circulation and produce ambient ocean temperature and salinity fields, we reduce model inputs to only basin-averaged values. Additionally, the geometry dependence of the plume model produces melt rates that show better agreement with observations in terms of both spatial distribution and magnitude than with PICO alone. Ocean induced melting has been cited as a major driver of change for Antarctic glaciers and over the coming century, enhanced spreading of mCDW onto the continental shelf is expected as Southern Ocean conditions are projected to change (Dinniman et al., 2012). As such, the improvements to the spatial distribution and magnitude of modeled melt rates produced by PICOP, as well as the computational efficiency of this parameterization, offer a valuable tool to more accurately make future projections of Antarctic glaciers.

*Code and data availability.*   The data used in this study are freely available on the National Snow and Ice Data Center, or upon request to the authors. ISSM is open source and freely available at http://issm.jpl.nasa.gov.

*Author contributions.*   TP developed the idea of combining PICO and a plume model, and implemented it into ISSM with help from MM and JB. All authors participated in the writing of the manuscript.

*Competing interests.*   None

*Acknowledgements.*   This work was performed at the University of California Irvine under a contract with the National Aeronautics and Space Administration, Cryospheric Sciences Program (#NNX15AD55G).

**Table 1.** Constant parameters and external quantities referenced in this communication. Common parameters are those used in the derivation of both PICO and the plume model. Unique parameters are from the derivation of the plume model, except for the overturning strength, which was taken from PICO. See Reese et al. (2018) and Lazeroms et al. (2018) for a full list of constants used to derive PICOP.

| External Quantity | Symbol | Source | Unit |
|---|---|---|---|
| Far-field ocean temperature | $T$ | Reese et al. (2018) | °C |
| Far-field ocean salinity | $S$ | Reese et al. (2018) | psu |
| Local depth of ice shelf base | $z_b$ | Fretwell et al. (2013) | m |
| Local slope angle | $\alpha$ | Geometry from Fretwell et al. (2013) | - |
| Grounding line depth | $z_{gl}$ | Solve advection problem | m |
| Ambient ocean temperature | $T_a$ | PICO | °C |
| Ambient ocean salinity | $S_a$ | PICO | psu |
| Basal melt rate | $\dot{m}$ | Plume model | m/yr |

| Common Constant Parameters | Symbol | Value | Unit |
|---|---|---|---|
| Gravitational acceleration | $g$ | 9.8 | m s$^{-2}$ |
| Density of ice | $\rho_i$ | 910 | kg m$^{-3}$ |
| Density of sea water | $\rho_w$ | 1028 | kg m$^{-3}$ |
| Latent heat of fusion | $L$ | $3.34 \cdot 10^5$ | J kg$^{-1}$ |
| Heat capacity of sea water | $c_p$ | 3974 | J kg$^{-1}$ °C |

| Unique Constant Parameters | Symbol | Value | Unit |
|---|---|---|---|
| Overturning strength | $C$ | $1 \cdot 10^6$ | m$^6$ s$^{-1}$kg$^{-1}$ |
| Entrainment coefficient | $E_0$ | $3.6 \cdot 10^{-2}$ | - |
| Drag coefficient | $C_d$ | $2.5 \cdot 10^{-3}$ | - |
| Turbulent heat exchange coefficient | $C_d^{1/2}\Gamma_T$ | $1.1 \cdot 10^{-3}$ | - |
| Freezing point-salinity coefficient | $\lambda_1$ | $-5.73 \cdot 10^{-2}$ | °C |
| Freezing point offset | $\lambda_2$ | $8.32 \cdot 10^{-2}$ | °C |
| Freezing point-depth coefficient | $\lambda_3$ | $7.61 \cdot 10^{-4}$ | °C m$^{-1}$ |
| Melt-rate parameter | $M_0$ | 10 | m yr$^{-1}$ °C$^{-2}$ |
| Heat exchange parameter | $C_d^{1/2}\Gamma_{TS_0}$ | $6.0 \cdot 10^{-4}$ | - |
| Heat exchange parameter | $\gamma_1$ | 0.545 | - |
| Heat exchange parameter | $\gamma_2$ | $3.5 \cdot 10^{-5}$ | m$^{-1}$ |
| Dimensionless scaling factor | $x_0$ | 0.56 | - |
| Epsilon | $\epsilon$ | $10^{-14}$ | - |

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
