# Peer review of "Brief communication: PICOP, a new ocean melt parameterization under ice shelves combining PICO and a plume model"

_The Cryosphere, 2018_

## Referee Comment (RC1) · Gwyther (Referee) · 29 Nov 2018

In this brief communication "PICOP, a new ocean melt parameterization under ice shelves combining PICO and a plume model", the authors Pelle et al, introduce a method for parameterising basal melting which will very likely prove useful in ice sheet models. An existing basal melting parametisation (PICO; Reese et al) is coupled to an existing plume model (Lazeroms et al), to produce estimates of basal melting that better consider ocean dynamics.

This manuscript presents a nice evolution in technique that I expect will prove useful. As such, I believe it is suitable for a brief communication-type paper in The Cryosphere.

[Figure]

My main concerns with this paper are that it properly recognises the original cavity and plume models that it couples. While this recognition occurs throughout the paper, it would be better suited right at the start, such as in the abstract. Secondly, comparison is made to glaciological/satellite-inferred estimates, which is fine, but observations and regional models also exist and would serve just as well for comparison. Thirdly, and the biggest issue in my opinion is how the ice velocity in the calculation of the plume. I believe the physical basis for this choice requires more justification.

I have several minor specific issues which I believe should be addressed. If these are addressed, I would recommend this manuscript for publication.

For reference, Page 5, Line 16 to 18 is referred to as P5L16-18.

Kind regards,

David Gwyther

University of Tasmania

Australia

Specific issues:

~~

Proper recognition: This paper is presenting a coupling procedure between existing models. As such, and considering that the authors of the original models are not co-authors, I think effort should be made to properly recognise the original model authors. One way to do this would be to cite the original model studies in the abstract. I note that both models are properly cited throughout the paper, but doing this early and upfront will better recognise the original studies.

Comparison to R2013: For specific locations, comparison should be done to actual observations, not just glaciological/satellite-inferred estimates. You refer to them as "observations", but there are existing observations (e.g. APRES) which resolve basal

melting better than a continent wide estimate/survey. Even though this is a brief communication, it will not take much space to compare to existing studies. For example,

Davis et al, 2018, "Variability in basal melting beneath Pine Island Ice Shelf on Weekly to Monthly Timescales"

Stanton et al., 2013 "Channelized ice melting in the ocean boundary layer beneath Pine Island Glacier, Antarctica"

Jenkins et al., 2010, "Observation and Parameterization of Ablation at the base of Ronne Ice Shelf, Antarctica"

Likewise, there are many regional modelling studies which possibly provide more consistent estimates for comparison with. For example, many studies exist for your target ice shelves, for Totten these are:

Khazendar et al., 2013, Observed thinning of Totten Glacier is linked to coastal polynya variability

Gwyther et al., 2014 Simulated melt rates for the Totten and Dalton ice shelves

Plumes coincide with ice velocities: Sometimes it's necessary that in order to overcome technical limitations, decisions must be made which may not be realistically accurate; this is a parametisation after all. However, there should be some realistic basis to coinciding plumes with high ice velocities. I think this should be explained more in depth. For example, it should be made clear whether the relatively good agreement with Rignot 2013 is because you expect actual plumes (and hence the high velocities and turbulence which drive melt rate) to be located in the region that your parameterisation predicts. That stability can not be achieved with plumes being initiated with high basal slope is disappointing; perhaps other initiation methods you tested could also be mentioned. Essentially what I'm saying is: are you getting the right answer for the wrong reasons, and in which case, would you expect these results to hold for future scenarios with evolving cavity geometry and a different ice velocity. This choice requires more justification and physical reasoning, including when this assumption could be expected to break down.

Technical corrections:

~~~

P1L8: capitalisation of Plume is not necessary.

P1L10: 'a wide variety' -> 'several'. You only showed us 3 regions.

P1L12: 'able to reproduce'. What you're referring to is 'able to match Rignot et al., 2013', which I think could be more accurately stated by saying 'able to reproduce inferred high melt rates beneath'...

P1L21: periphery OF the AIS

P1L21-P2L1: long sentence, consider splitting in half.

P7L2: "better agreement with observation than PICO, because PICOP..."

P7L19-23: Sentence is way too long; split in 2 or 3.

Figure 1: "on" in green box has mismatched font.

Table 1: in the External Quantity section, you have many "-" in the value column. I would change the column header to "Source", and then cite the study where you pull the quantities from e.g. Far-field ocean temperature, etc.

Table 1: missing '\epsilon' value?

---

## Referee Comment (RC2) · Hellmer (Referee) · 19 Dec 2018

In this brief communication, the authors present a new parameterization of ocean induced ice shelf basal melting based on a quite simple numerical model. PICOP is a combination of the simplified box model (PICO) and a 2-d plume model. The authors claim that the model is more appropriate to use for transient coupled ice sheet – ocean model runs, since it resolves many of the problems caused by the simplifications/ assumptions made in the original models.

General comments:

A Brief Communication in TC might allow for a more superficial description of the various model components, but to gain a thorough insight the reader is forced to read the original publications of Reese et al. (2018) and Lazeroms et al. (2018). Without doing so, one has to trust that the model set-up is correct, the right parameters have been used, and a realistic forcing is applied. Thereby, the only way left for evaluation is the comparison of basal melt rates (Fig. 5) for three different ice shelf regions at the rim of the Antarctic Ice Sheet. The latter, however, bears some subjectivity and might have tempted the authors to use the term 'in excellent agreement'. However, first, the authors 'only' compare their results with one observation (Rignot et al., 2013), which actually is not an in-situ observation. Second, it is not obvious to the reader which set of parameters has be used, what tuning has been applied to reach this agreement. Third, the agreement for Filchner-Ronne Ice Shelf (FRIS) is difficult to assess since an inappropriate color scale is used. Various approaches show, and the results of Reese et al. (2018) greatly exaggerate, that refreezing occurs below FRIS. I hope it is only a matter of the color scale, since the pattern shows promising. However, melting at the ice shelf front can also not be resolved by PICOP, since it is a different process which drives this frontal melting.

Therefore, I urge the authors to spend more time on validation of the PICOP results, but if done, I recommend publication in TC after consideration of the comments/ corrections listed below.

Specific comments:

P1L18: The term 'upwelling' is not correctly used. In Physical Oceanography, 'upwelling' means the vertical displacement of deep water masses towards the surface, caused by the prevailing winds, e.g., at the Antarctic Divergence, off Namibia, etc. Here, it is the spreading of mCDW onto the continental shelf and, sometimes, into the fringing ice shelf cavities.

P2L16: The name might be misleading, but PICO was NOT designed to reproduce the

density driven overturning circulation within sub-ice shelf cavities. The latter, however, was the intention of the box model developed by Olbers & Hellmer (2010).

P4L03: It is unclear where the ice depth-averaged velocity 'v' is actually coming from – either from the ice-sheet model coupled to PICOP or from observations. In addition, 'epsilon' is not listed in Table 1.

P4L07: Same for $z\_b$ – assumed to be taken from Bedmap2.

P4L27: The assumption that the ambient temperature in the ice shelf cavity $T\_a$ is equal or above the surface freezing point is unfortunate, at least for cold cavities like FRIS and Ross Ice Shelf, since the observations show temperatures well below $T\_f$ characteristic of Ice Shelf Water. This assumption might lead to an exaggeration of basal melting in these cavities.

P5L12: The mean salinity for the Amundsen Sea of 34.86 is wrong, e.g., Dutrieux et al. (2014), and seems to be a transposed digits since 34.68 makes more sense. Such high salinity causes a stronger overturning circulation and thus higher basal melting. Same for the southern Weddell Sea continental shelf, where a salinity above 34.8 is only observed in the far western corner, which – I have to admit – provides most of the shelf water fueling basal melting underneath Ronne Ice Shelf.

Table 1: Checking the constant parameters used in Reese et al. (2018) one realizes that they sometimes differ from those used for PICOP, which leaves the impression that Table 1 shows a mixture of parameters used in the original models. In addition, the overturning strength C used in Reese et al. (2018) – Table 1 was a best-fit, which does not necessarily mean that a strength of 1x10ˆ6 is also the best choice for PI-COP. Finally, please explain why most units include '°C' but the 'freezing point-depth coefficient'.

Technical corrections

P1L01: 'basal melt' is the liquid resulting from ocean induced ice shelf 'basal melting'

– please use the latter term.

P1L11 - and the following: The term Filchner-Ronne Ice Shelf is widely recognized, e.g. in Reese et al. (2018).

P1L21: "... along the periphery OF the AIS....; sentence too long, split in two.

P3L08: Temperature 'T' and salinity 'S' are already used for the far field temperature and salinity. I suggest omitting "with temperature T, and salinity S".

P7L21: "... are projected to change within the coming century, ... " needs a reference.

References

Olbers, D. and Hellmer, H. H. (2010) A box model of circulation and melting in ice shelf caverns. Ocean Dynamics, 60, 141-153, https://doi.org/10.1007/s10236-009-0252-z.

Dutrieux et al. (2014) Strong sensitivity of Pine Island Ice Shelf melting to climate variability. Science, 343, 174-178, https://doi.org/10.1126/science.1244341

---

## Author Comment (AC1) · 25 Jan 2019

**Brief communication: PICOP, a new ocean melt parameterization under ice shelves combining PICO and a plume model**
**– Response to reviewers –**

Tyler PELLE et al.

January 24, 2019

We thank the two reviewers for their positive and constructive comments that significantly improved the manuscript. While reviewing the manuscript, we found that we used the wrong temperature and salinity input for the Amundsen Sea sector. The correct and updated input for this sector is T $= 0.47^{\circ}$C and S $= 34.73$ psu. As such, we ran PICOP with the correct input and updated the figures and text accordingly. In Fig. 1, we updated the Pine Island ice shelf graphics that show the blank ice shelf with the input, $T_a$, $S_a$, and $\dot{m}$. In Fig. 2, we updated the basal melt rate fields for PICO and PICOP, as well as the shelf-wide means. The pattern of basal melting near the grounding lines of Pine Island and Thwaites glaciers remained the same; however, melt rates away from the grounding line decreased significantly as a result of decreased overturning. In addition to making this change, we also address the reviewer's remarks below point by point.

**1   Reviewer #1**

*Proper recognition: This paper is presenting a coupling procedure between existing models. As such, and considering that the authors of the original models are not coauthors, I think effort should be made to properly recognize the original model authors. One way to do this would be to cite the original model studies in the abstract. I note that both models are properly cited throughout the paper, but doing this early and upfront will better recognize the original studies.*

We agree with the reviewer and have cited both published models in the abstract of the manuscript.

*Comparison to R2013: For specific locations, comparison should be done to actual observations, not just glaciological/ satellite-inferred estimates. You refer to them as "observations", but there are existing observations (e.g. APRES) which resolve basal melting better than a continent wide estimate/survey. Even though this is a brief communication, it will not take much space to compare to existing studies.*

This is a great point and we agree that our validation of PICOP would benefit from comparison to existing studies. We added additional comparisons in the discussion section.

*Plumes coincide with ice velocities: Sometimes it's necessary that in order to overcome technical limitations, decisions must be made which may not be realistically accurate; this is a parameti-sation after all. However, there should be some realistic basis to coinciding plumes with high ice velocities. I think this should be explained more in depth. For example, it should be made clear whether the relatively good agreement with Rignot 2013 is because you expect actual plumes (and hence the high velocities and turbulence which drive melt rate) to be located in the region that your parameterisation predicts. That stability can not be achieved with plumes being initiated with high basal slope is disappointing; perhaps other initiation methods you tested could also be mentioned. Essentially what I'm saying is: are you getting the right answer for the wrong reasons, and in which case, would you expect these results to hold for future scenarios with evolving cavity geometry and a different ice velocity. This choice requires more justification and physical reasoning, including when this assumption could be expected to break down.*

Physical justification and clarification of the choice to use ice velocities to advect grounding-line heights has been added to the PICOP-methods section. A description of when this assumption is expected to break down has been added to the discussion section.

*Technical corrections:*
*P1L8: capitalisation of Plume is not necessary.*

Done.

*P1L10: "a wide variety" change to "several." You only showed us 3 regions.*

Done.

*P1L12: "able to reproduce". What you're referring to is "able to match Rignot et al., 2013", which I think could be more accurately stated by saying "able to reproduce inferred high melt rates beneath"...*

We agree and changed the wording of this sentence.

 *P1L21: periphery OF the AIS*

56 Done.

57 *P1L21-P2L1: long sentence, consider splitting in half.*

58 Done.

59 *P7L2: "better agreement with observation than PICO, because PICOP."*

60 Done.

61 *Figure 1: "on" in green box has mismatched font.*

62 We changed the font in the green box of Fig. 1 to match the other boxes.

63 *Table 1: in the External Quantity section, you have many "-" in the value column. I would change*
64 *the column header to "Source", and then cite the study where you pull the quantities from e.g.*
65 *Far-field ocean temperature, etc.*

66 Done. For quantities that are computed by PICOP from the input data, we specify which model
67 component calculates it (either PICO or the plume model).

68 *Table 1: missing $\epsilon$ value?*

69 We added $\epsilon$ to Table 1.

**2   Reviewer #2**

71 *A Brief Communication in TC might allow for a more superficial description of the various model*
72 *components, but to gain a thorough insight the reader is forced to read the original publications of*
73 *Reese et al. (2018) and Lazeroms et al. (2018). Without doing so, one has to trust that the model*
74 *set-up is correct, the right parameters have been used, and a realistic forcing is applied.*

75 We believe the text presented is suitable as a Brief Communication in TC because we couple two
76 existing, published models. Both models were implemented into ISSM without alteration (except
77 for the computation of the grounding line height in the plume model, which is described in the

methods section), so we do not think it is necessary to restate the complete setup of both models. Rather, the methods section provides an overview of the two models, guides the reader to the appropriate sections of the original publications, and details the coupling between the two original models. Forcing applied to PICOP was taken from Reese et al. (2018) to facilitate comparison between PICO and PICOP.

*The only way left for evaluation is the comparison of basal melt rates (Fig. 5) for three different ice shelf regions at the rim of the Antarctic Ice Sheet. The latter, however, bears some subjectivity and might have tempted the authors to use the term "in excellent agreement". However, first, the authors "only" compare their results with one observation (Rignot et al., 2013), which actually is not an in-situ observation.*

We agree that the validation of PICOP would benefit from additional comparison, aside from Rignot et al. (2013). We added more comparisons in the discussion section of the text, including high definition ocean modeling studies and in-situ observations.

*It is not obvious to the reader which set of parameters has be used, what tuning has been applied to reach this agreement.*

Since both models were implemented without change from the original publications, the full list of parameters that were used to implement PICOP are found in the original publications. There was no tuning applied to PICOP; the only quantities that changed in the computation of the basal melt rate fields for the three basins were the far field ocean temperature and salinity (taken from Reese et al. (2018)).

*The agreement for Filchner-Ronne Ice Shelf (FRIS) is difficult to assess since an inappropriate color scale is used. Various approaches show, and the results of Reese et al. (2018) greatly exaggerate, that refreezing occurs below FRIS. I hope it is only a matter of the color scale, since the pattern shows promising.*

We agree that the color scale used in Fig. 2 for the Filchner-Ronne Ice Shelf (FRIS) was not appropriate, as it did not show refreezing and was also over-saturated. We changed the color scale for the FRIS from log to linear, changed the color map to reduce saturation, and extended the bounds to include refreezing in gray-scale. We note that PICOP does not model refreezing well due to the lower bound applied to the ambient ocean temperature and highlight it as one of the limitations of PICOP in the discussion section.

*Melting at the ice shelf front can also not be resolved by PICOP, since it is a different process which drives this frontal melting.*

The reviewer is correct, we removed this sentence from the text that incorrectly described the frontal melt of the FRIS.

*Specific comments:*

*P1L18: The term "upwelling" is not correctly used. In Physical Oceanography, "upwelling" means the vertical displacement of deep water masses towards the surface, caused by the prevailing winds, e.g., at the Antarctic Divergence, off Namibia, etc. Here, it is the spreading of mCDW onto the continental shelf and, sometimes, into the fringing ice shelf cavities.*

This is a good point and we changed the manuscript to correctly describe this process.

*P2L16: The name might be misleading, but PICO was NOT designed to reproduce the density driven overturning circulation within sub-ice shelf cavities. The latter, however, was the intention of the box model developed by Olbers and Hellmer (2010).*

We changed the description to more correctly say that PICO simulates vertical overturning, as was stated in Reese et al. (2018).

*P4L03: It is unclear where the ice depth-averaged velocity $v$ is actually coming from – either from the ice-sheet model coupled to PICOP or from observations. In addition, $\epsilon$ is not listed in Table 1.*

We added clarification at this point in the text; the depth-averaged velocity is taken from the ice-sheet model. We also added $\epsilon$ to Table 1.

*P4L07: Same for $z_b$ assumed to be taken from Bedmap2.*

We added the source of input data to Table 1, which will clarify this to the reader. We also state that we use geometry from Bedmap2 at the beginning of the results section.

*P4L27: The assumption that the ambient temperature in the ice shelf cavity $T_a$ is equal or above the surface freezing point is unfortunate, at least for cold cavities like FRIS and Ross Ice Shelf, since the observations show temperatures well below $T_f$ characteristic of Ice Shelf Water. This assumption might lead to an exaggeration of basal melting in these cavities.*

We highlight the lower bound applied to the ambient ocean temperature as a limitation of PICOP in the discussion section. We also added a statement about how cold-cavity basal melt rates may be over estimated by PICOP and reference FRIS in Fig. 2.

*P5L12: The mean salinity for the Amundsen Sea of 34.86 is wrong, e.g., Dutrieux et al. (2014) and seems to be a transposed digits since 34.68 makes more sense. Such high salinity causes a*

*stronger overturning circulation and thus higher basal melting. Same for the southern Weddell Sea continental shelf, where a salinity above 34.8 is only observed in the far western corner, which, I have to admit, provides most of the shelf water fueling basal melting underneath Ronne Ice Shelf.*

We recognize that these mean salinity values may be in slight disagreement with observations (Dutrieux et al. (2014) reports a mean salinity that is ∼0.2 psu lower than our value used for the Amunsen Sea sector). The mean temperature and salinity values used to force PICOP were taken directly from Reese et al. (2018) and we use them for the sake of comparison. In addition, we ran PICOP with a mean salinity of 34.68 psu over the Amundsen Sea sector and found that the small deviation in mean salinity did not have a significant impact on the modeled basal melt rates.

*Table 1: Checking the constant parameters used in Reese et al. (2018) one realizes that they sometimes differ from those used for PICOP, which leaves the impression that Table 1 shows a mixture of parameters used in the original models. In addition, the overturning strength C used in Reese et al. (2018) – Table 1 was a best-fit, which does not necessarily mean that a strength of $1 \times 10^6$ is also the best choice for PICOP. Finally, please explain why most units include °C but the "freezing point-depth coefficient."*

We reorganized Table 1 to group constant parameters that are common to both of the original publications and those that are unique to each publication. We recognize that the overturning strength is a best-fit value for PICO and there may be a more appropriate value for PICOP. However, we use $1 \times 10^6$ in our study for the sake of comparison between PICOP and PICO. Lastly, we changed the units for the freezing point-depth coefficient to include °C rather than K.

*Technical corrections:*
*P1L01: "basal melt" is the liquid resulting from ocean induced ice shelf "basal melting" – please use the latter term.*

Done.

*P1L11 - and the following: The term Filchner-Ronne Ice Shelf is widely recognized, e.g. in Reese et al. (2018).*

We changed the name of this ice shelf from Ronne-Filchner to Filchner-Ronne in the text.

*P1L21: "... along the periphery OF the AIS...."; sentence too long, split in two.*

Done.

*P3L08: Temperature T and salinity S are already used for the far field temperature and salinity. I suggest omitting "with temperature T, and salinity S".*

170 Done.

171 *P7L21: "… are projected to change within the coming century, … " needs a reference.*

172 Done.

**References**

Dutrieux, P., De Rydt, J., 1 Jenkins, A., Holland, P., H.K., H., Lee, S., Steig, E., Ding, Q., Abrahamsen, E., and Schröder, M.: Strong Sensitivity of Pine Island Ice Shelf Melting to Climatic Variability, Science, 343, 174–178, https://doi.org/10.1126/science.1244341, 2014.

[revised manuscript text omitted]

**References**

[revised manuscript text omitted]